# REV3 promotes cellular tolerance to 5-fluorodeoxyuridine by activating translesion DNA synthesis and intra-S checkpoint

Mubasshir Washif[☉], Ryotaro Kawasumi[☉], Kouji Hirota[☉]*

Department of Chemistry, Graduate School of Science, Tokyo Metropolitan University, Tokyo, Japan

☉ These authors contributed equally to this work.
* khirota@tmu.ac.jp

**Data Availability Statement:** The authors confirm that all data underlying the findings are fully available without restriction. All relevant data are

## Abstract

The drug floxuridine (5-fluorodeoxyuridine, FUdR) is an active metabolite of 5-Fluorouracil (5-FU). It converts to 5-fluorodeoxyuridine monophosphate (FdUMP) and 5-fluorodeoxyuridine triphosphate (FdUTP), which on incorporation into the genome inhibits DNA replication. Additionally, it inhibits thymidylate synthase, causing dTMP shortage while increasing dUMP availability, which induces uracil incorporation into the genome. However, the mechanisms underlying cellular tolerance to FUdR are yet to be fully elucidated. In this study, we explored the mechanisms underlying cellular resistance to FUdR by screening for FUdR hypersensitive mutants from a collection of DT40 mutants deficient in each genomic maintenance system. We identified REV3, which is involved in translesion DNA synthesis (TLS), to be a critical factor in FUdR tolerance. Replication using a FUdR-damaged template was attenuated in $REV3^{-/-}$ cells, indicating that the TLS function of REV3 is required to maintain replication on the FUdR-damaged template. Notably, FUdR-exposed $REV3^{-/-}$ cells exhibited defective cell cycle arrest in the early S phase, suggesting that REV3 is involved in intra-S checkpoint activation. Furthermore, $REV3^{-/-}$ cells showed defects in Chk1 phosphorylation, which is required for checkpoint activation, but the survival of FUdR-exposed $REV3^{-/-}$ cells was further reduced by the inhibition of Chk1 or ATR. These data indicate that REV3 mediates DNA checkpoint activation at least through Chk1 phosphorylation, but this signal acts in parallel with ATR-Chk1 DNA damage checkpoint pathway. Collectively, we reveal a previously unappreciated role of REV3 in FUdR tolerance.

## Author summary

Nucleoside analogs have been frequently used for the treatment of virus infections and cancers. These drugs restrict the proliferation of viruses and cancers by interfering with DNA replication. The drug floxuridine (5-fluorodeoxyuridine, FUdR) is a thymidine analog and used for treating hepatic metastases of colorectal cancer. However, the effect of this drug on healthy non-cancer cells and the mechanisms underlying the cellular tolerance to FUdR remain elusive. We here determined the cellular effect of FUdR and the repair system involved the cellular tolerance to FUdR. We found that REV3, a polymerase

within the paper and its Supporting Information files.

**Funding:** Financial support was provided by grants from JSPS KAKENHI (JP21K19235, JP20H04337, and 19KK0210), Tokyo Metropolitan Government Advanced Research Grant Number (R3-2), Takeda Science Foundation, and Yamada Science Foundation for KH, and JSPS KAKENHI (JP22K15040) for RK. The funders had no role in study design, data collection and analysis, decision to publish, or preparation of the manuscript.

**Competing interests:** The authors have declared that no competing interests exist.

involved in translesion DNA synthesis (TLS), promotes bypass replication on a FUdR-damaged template. Moreover, REV3 promotes DNA checkpoint activation and induces cell-cycle arrest in the early S-phase, which is required for the suppression of FUdR incorporation in the genome thereby contributing to the suppression of chromosome instability. In this study, we uncovered the previously unappreciated roles of REV3; the promotion of the cellular tolerance to FUdR by activating TLS and intra-S checkpoint.

## Introduction

Nucleoside analogs are chemical compounds that are similar in structure to nucleosides. They have been used as anti-cancer and anti-viral agents [1,2] as the incorporation of these drugs into genomic DNA interferes with DNA replication, which effectively restricts the proliferation of cancers and viruses [3]. Thus far, nucleoside analogs are known to interfere with DNA replication in three ways [4,5]: they are incorporated at the end of nascent DNA and subsequently interfere with polymerization reactions by acting as chain terminators; they are incorporated into the genome and restrict DNA replication on the nucleoside analog-incorporated template; they inhibit metabolism essential for nucleotide generation, which affects DNA replication.

Several DNA repair systems maintain cellular tolerance to the inhibitory effects of nucleoside analogs on DNA replication [6,7]. For example, exonuclease activity of polymerase epsilon and tyrosyl-DNA-phosphodiesterase 1 (TDP1) removes the inserted nucleoside analogs from the end of nascent DNA to mitigate chain-terminating effects [4,8,9]. Translesion DNA synthesis (TLS) and homologous recombination (HR) processes release the arrested replication fork on the nucleoside analog-incorporated damaged template [10–12]. TLS involves specialized DNA polymerases called TLS polymerases that synthesize DNA using damaged templates [13]. Polymerase ζ (Polζ) is a type of TLS polymerase that possesses REV3 as a catalytic subunit and requires REV1 and REV7 for its TLS activity [14,15]. In the current model, Y-family polymerases including Polymerase η (Polη) insert nucleotide opposite the damaged site, and Polζ extends from the mismatched primer end to complete TLS [14,16,17]. HR-mediated template switching also contributes to the release of stalled replication at the damaged site. In this mechanism, stalled replication at the damaged site is released by switching the template strand from a damaged to a nascent damage-free template [18]. Replication arrest results in gapped DNA where the single-stranded DNA is exposed, which activates DNA damage checkpoints. RAD17 is a component of the alternative clamp loader RAD17-RFC, which loads the 9-1-1 checkpoint clamp. This stimulates the DNA damage checkpoint by the induction of Chk1 phosphorylation [7,19,20]. Additionally, ATR contributes to DNA damage sensing and activates the DNA damage checkpoint via Chk1 phosphorylation [21]. Moreover, replication arrest, which results in replication fork collapse, induces DNA double-strand breaks (DSBs) [22]. These DSBs are repaired by HR and non-homologous end joining (NHEJ) processes—HR repairs DSB in an error-free manner, whereas NHEJ induces insertions and deletions during repair [23,24]. Base excision repair (BER) is another mechanism that removes the incorporated nucleoside analogs, in which PARP1 acts as a DNA damage sensor [25,26]. However, the repair pathways required for cellular tolerance to nucleoside analogs vary distinctly depending on the drug, and the mechanisms underlying these differences have not yet been elucidated.

Floxuridine (5-fluorodeoxyuridine, FUdR) is an active metabolite of 5-fluorouracil (5-FU). FUdR has been approved by the U.S. Food and Drug Administration for treating hepatic metastases of colorectal cancer and other gastrointestinal tract tumors [27]. Despite FUdR

being a metabolite of 5-FU, both compounds exert different mechanisms of action in human tumor cells [5]. On digestion, FUdR is metabolized by cellular thymidine kinase into two active metabolites, namely, 5-fluoro-2-deoxyuridine monophosphate (5-FdUMP) and 5-fluoro-2-deoxyuridine triphosphate (5-FdUTP). FUdR primarily induces cell death by altering DNA metabolism by inhibiting thymidine synthase activity, which depletes the cellular dTTP pool and promotes dUTP accumulation. During replication, dUMP and 5-FdUMP are incorporated into the genomic DNA, which increases the dUMP and 5-FdUMP levels in the genome. Both outcomes, namely, the imbalance in the cellular dNTP pool and the misincorporation of dUMP and 5-FdUMP into the genome, cause the activation of ATR-dependent intra-S checkpoint signaling [28–33]. DNA damage induced by the misincorporation of dUMP and 5-FdUMP into the genome is targeted by several DNA repair pathways including the mismatch repair and BER pathways [5,28–31]. Moreover, the involvement of HR in cellular tolerance to FUdR has been previously reported [34]. However, the repair mechanisms underlying cellular tolerance to FUdR are yet to be fully elucidated.

In this study, we aimed to identify repair pathways by screening DT40 mutant cells deficient in each genome maintenance system to identify mutants showing hypersensitivity to FUdR. During the screening, we detected the factors involved in the previously identified pathways required for FUdR tolerance (in BER, checkpoint signaling, and HR processes) [28–33]. Additionally, we identified REV3 as a critical factor for FUdR tolerance. REV3 promotes FUdR tolerance through activating TLS and checkpoint. Taken together, these findings provide experimental clarity regarding REV3-mediated cellular tolerance to FUdR via TLS and checkpoint activation.

## Results

### Screening of DNA repair mutants showing hypersensitivity to FUdR

To investigate the mechanisms underlying cellular tolerance against FUdR-induced DNA damage, we explored mutant cell lines showing FUdR-hypersensitive phenotype from our mutant cell line collection generated from the chicken DT40 cells, including mutants deficient in TLS ($REV3^{-/-}$, $RAD18^{-/-}$, $POLH^{-/-}$, $PRIMPOL^{-/-}$ $SPRTN^{-/-}$) [12,14,35–37], HR ($BRCA1^{-/-}$, $BRCA2^{-/-}$) [38], NHEJ ($POLQ^{-/-}$, $KU70^{-/-}$) [39], HR and NHEJ ($RAD54^{-/-}$ $KU70^{-/-}$) [40,41], checkpoint signaling ($ATM^{-/-}$, $RAD17^{-/-}$) [42,43], BER ($PARP1^{-/-}$, $FEN1^{-/-}$, $POLB^{-/-}$) [44–46], nucleotide excision repair (NER; $XPA^{-/-}$) [14], the Fanconi Anemia pathway (FA; $FANCC^{-/-}$, $FANCJ^{-/-}$) [47,48], proofreading ($POLE1^{D269A/-}$) [4], removal of topoisomerase I-DNA cleavage complexes (Top1-cc repair; $TDP1^{-/-}$ $TDP2^{-/-}$) [49], and sister chromatid cohesion (SCC; $DDX11^{-/-}$, $CTF18^{-/-}$, $SA2^{-/-}$) [50–52] (Table 1). We assessed cellular sensitivity to FUdR using the ATP assay, and from these sensitivity data, we calculated the relative sensitivity using the formula $\log_2$ ($LD_{90}$ in the indicated mutant cells)/($LD_{90}$ in the wild-type cells) (Fig 1A), where $LD_{90}$ indicates the drug concentration that reduces cellular survival by 10%. We assessed the mutants showing increased sensitivity to FUdR (relative sensitivity $< -2$) and weak or no sensitivity (relative sensitivity $> -2$) (Figs 1A and S1). In conformance with previous reports [28–33], we found that mutant cells deficient in HR ($BRCA1^{-/-}$ and $BRCA2^{-/-}$), BER ($PARP1^{-/-}$), and checkpoint signaling ($RAD17^{-/-}$) showed higher sensitivity to FUdR than wild-type cells. However, with respect to pathways other than those known to be responsible for FUdR tolerance, $REV3^{-/-}$ cells showed the highest FUdR sensitivity among the tested mutants (Fig 1A). $REV3^{-/-}$ human TK6 cells also showed increased FUdR sensitivity (Fig 1B). These data suggest that besides HR, BER, and checkpoint signaling, TLS also plays crucial roles in cellular tolerance to FUdR. To test the involvement of the TLS pathway in FUdR tolerance, we assessed the cellular sensitivity of other TLS-deficient mutants including $REV1^{-/-}$ [15], $POLD3^{-/-}$ [11], $POLH^{-/-}$ [14],

**Table 1. List of cell lines used in this study.**

| Genotype | Parental Cell line | Function | References |
|---|---|---|---|
| Wild-type | DT40 | | [75] |
| *BRCA1*$^{-/-}$ | Wild-type DT40 cells | HR | [76] |
| *BRCA2*$^{-/-}$ | Wild-type DT40 cells | HR | [77] |
| *POLQ*$^{-/-}$ | Wild-type DT40 cells | NHEJ | [46] |
| *KU70*$^{-/-}$ | Wild-type DT40 cells | NHEJ | [39] |
| *RAD54*$^{-/-}$ / *KU70*$^{-/-}$ | Wild-type DT40 cells | HR/NHEJ | [39] |
| *POLB*$^{-/-}$ | Wild-type DT40 cells | BER | [46] |
| *PARP1*$^{-/-}$ | Wild-type DT40 cells | BER | [71] |
| *FEN1*$^{-/-}$ | Wild-type DT40 cells | BER | [48] |
| *XPA*$^{-/-}$ | Wild-type DT40 cells | NER | [14] |
| *FANCC*$^{-/-}$ | Wild-type DT40 cells | Fanconi Anemia | [47] |
| *FANCJ*$^{-/-}$ | Wild-type DT40 cells | Fanconi Anemia | [48] |
| *SPRTN*$^{-/-}$ | Wild-type DT40 cells | TLS / Protein-DNA repair | [36] |
| *REV3*$^{-/-}$ | Wild-type DT40 cells | TLS | [10] |
| *RAD18*$^{-/-}$ | Wild-type DT40 cells | TLS | [35] |
| *POLH*$^{-/-}$ | Wild-type DT40 cells | TLS | [14] |
| *PRIMPOL*$^{-/-}$ | Wild-type DT40 cells | Repriming | [12] |
| *POLE1*$^{exo -/-}$ | Wild-type DT40 cells | Removal of nucleoside analogs | [4] |
| *TDP1*$^{-/-}$/*TDP2*$^{-/-}$ | Wild-type DT40 cells | Protein tyrosyl-DNA repair | [49] |
| *ATM*$^{-/-}$ | Wild-type DT40 cells | Checkpoint | [42] |
| *RAD17*$^{-/-}$ | Wild-type DT40 cells | Checkpoint | [43] |
| *DDX11*$^{-/-}$ | Wild-type DT40 cells | Cohesion | [50] |
| *CTF18*$^{-/-}$ | Wild-type DT40 cells | Removal of nucleoside analogs | [51] |
| *SA2*$^{-/-}$ | Wild-type DT40 cells | Cohesion | [52] |
| *POLK*$^{-/-}$ | Wild-type DT40 cells | TLS | [14] |
| *REV1*$^{-/-}$ | Wild-type DT40 cells | TLS | [15] |
| *POLD3*$^{-/-}$ | Wild-type DT40 cells | TLS | [11] |
| *PCNA*$^{-/K164R}$ | Wild-type DT40 cells | TLS | [53] |
| Wild-type | TK6 | | [78] |
| *REV3*$^{-/-}$ | Wild-type TK6 cells | TLS | [79] |

*POLK*$^{-/-}$ [14], *RAD18*$^{-/-}$ [35], and *PCNA*$^{-/K164R}$ [53,54](Fig 1C). *REV1*$^{-/-}$ cells showed increased sensitivity, comparable to that of *REV3*$^{-/-}$ cells, whereas other mutants showed moderate sensitivity, which was much lower than that of *REV1*$^{-/-}$ and *REV3*$^{-/-}$ cells (Fig 1C). These tendencies were also seen in human TK6 cells (Fig 1D). Thus, these data suggest that the TLS pathway is involved in cellular tolerance to FUdR.

## REV3 maintains replication fork progression on FUdR-damaged template

TLS process avoids replication fork arrest on damaged templates [13]. We hypothesized that if the TLS function of REV3 is responsible for FUdR tolerance, the replication fork on the 5-FdUMP- and dUMP-incorporated damaged templates would be attenuated more frequently in *REV3*$^{-/-}$ cells than in wild-type cells. We tested this hypothesis using a DNA fiber assay to monitor replication fork speed. To prepare 5-FdUMP- and dUMP-incorporated template strands, we cultured the cells in the presence of FUdR for 12 h (the duration of the whole cell cycle in DT40 is 8 h), released them into a drug-free medium for 30 min, and sequentially pulse-labeled them with CldU and IdU to assess replication dynamics (Fig 2A). As expected,

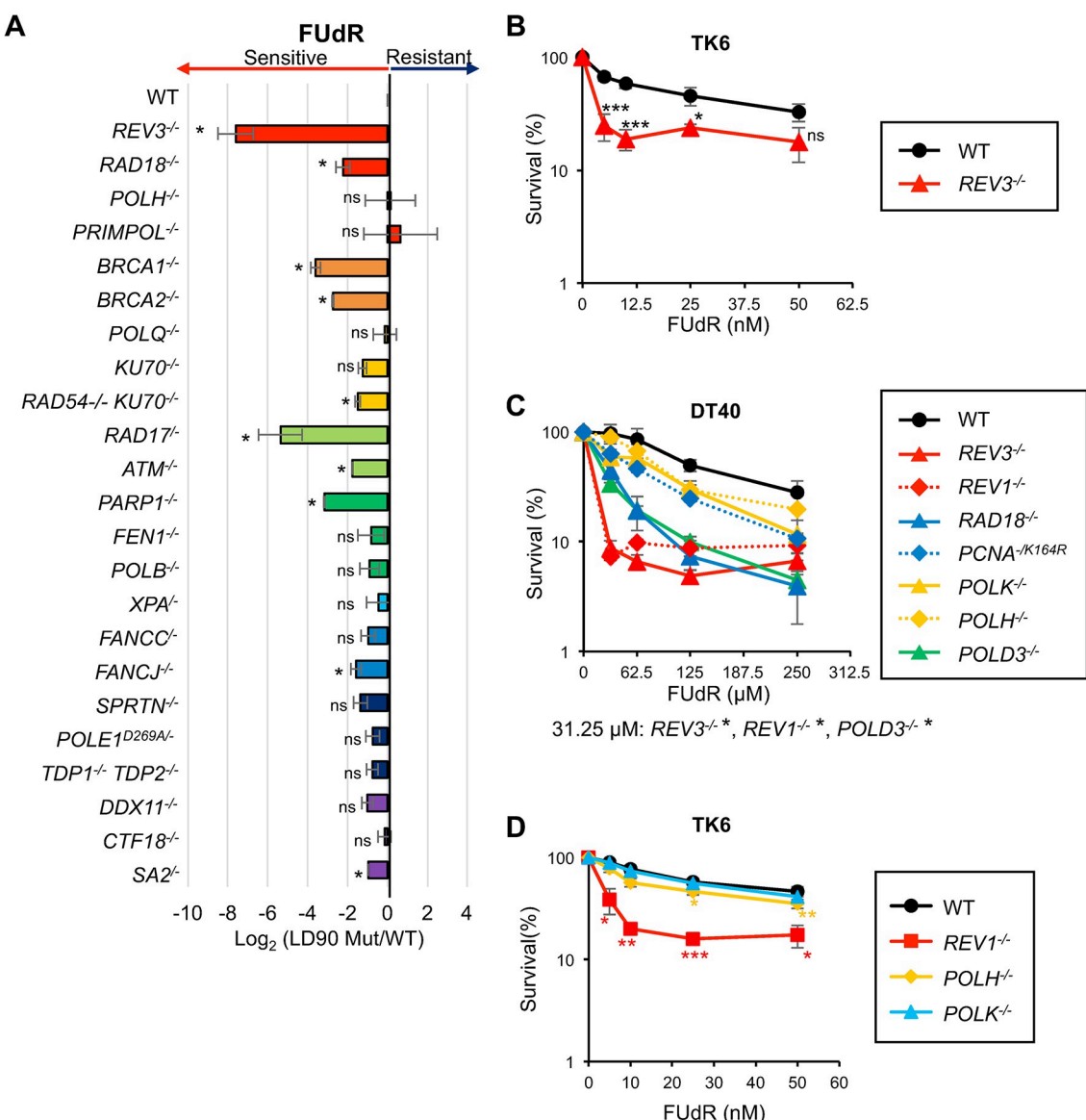

**Fig 1. Identification of REV3 as a critical factor for cellular tolerance to FUdR.** (A) Sensitivity profiles of FUdR in the selected DT40 mutant cells deficient in individual DNA damage response pathways. Sensitivity to FUdR in cells with indicated genotype was measured by ATP assay as described in Materials and Methods. The relative sensitivity of each mutant cell compared with that of wild-type DT40 cells was scored as log2 ($LD_{90}$ in indicated mutant cells)/($LD_{90}$ in wild-type cells), where $LD_{90}$ represents the drug concentration that reduces cell survival to 10% relative to that of untreated cells. Negative (left) and positive (right) scores show that the indicated gene-disrupted cells are sensitive and resistant to FUdR, respectively. (B) TK6 cells with the indicated genotypes were assessed for sensitivity to FUdR. Indicated TK6 cells were cultured for 14 days in the presence of an indicated concentration of FUdR. The dose of FUdR is displayed on the x-axis on a linear scale, whereas the percentage of cell survival is displayed on the y-axis on a logarithmic scale. Error bars represent the standard deviation from three independent experiments. (C) DT40 cells with the indicated genotypes were assessed for sensitivity to FUdR. Indicated DT40 cells were cultured for 48 h in the presence of an indicated concentration of FUdR. (D) TK6 cells with the indicated genotypes were assessed for sensitivity to FUdR. Indicated TK6 cells were cultured for 14 days in the presence of an indicated concentration of FUdR. Error bars represent the standard deviation from three independent experiments. *p*-values were calculated by student t-test. ns, not significant; *$p < 0.05$; **$p < 0.01$; ***$p < 0.001$.

the kinetics of the replication fork progression slowed when the template strand was damaged (Fig 2B and 2C). Notably, compared with that of wild-type cells, the rate of DNA synthesis in *REV3*-/- cells was significantly slower under perturbed conditions (Fig 2B and 2C), and we

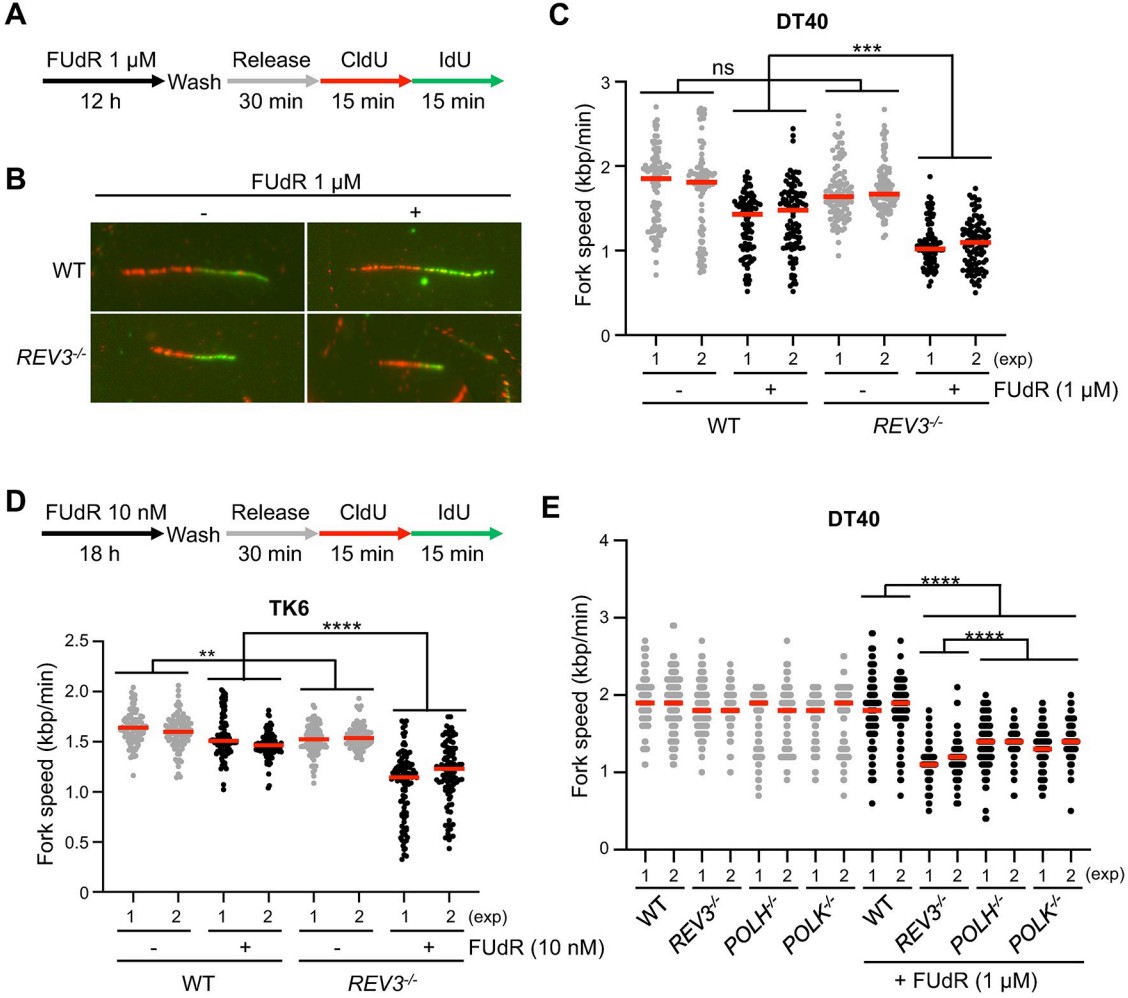

**Fig 2. REV3 maintains fork progression on the FUdR-damaged templates.** (A) Schematic representation of the experimental protocol to monitor fork speed. DT40 or TK6 cells were cultured in a medium containing 1 μM FUdR for 12 h, followed by release in a drug-free medium and culturing for 30 min. The cells were sequentially pulse-labeled with CldU and IdU for 15 min each. (B) Representative images showing DNA fibers from DT40 cells. (C) Quantification results of fork speed in DT40 cells. The red line indicates the median. All statistical analyses were performed using the Mann–Whitney–Wilcoxon test. ns, not significant; ***$p < 0.001$. (D) Quantification results of fork speed in TK6 cells. The red line indicates the median. All statistical analyses were performed using the Mann–Whitney–Wilcoxon test. ns, not significant; **$p < 0.01$; ****$p < 0.0001$. (E) Quantification results of fork speed in DT40 cells. The red line indicates the median. All statistical analyses were performed using the Mann–Whitney–Wilcoxon test. ****$p < 0.0001$.

observed similar results in human TK6 cells (Fig 2D). Moreover, $POLH^{-/-}$ and $POLK^{-/-}$ cells also showed slower fork progression in perturbed conditions than did wild-type cells (Fig 2E). These data indicate that TLS maintains replication fork progression in the presence of 5-FdUMP and dUMP on template strands.

## REV3 prevents DNA damage after FUdR incorporation in nascent DNA

As only the TLS factors associated with Polζ ($REV1^{-/-}$ and $REV3^{-/-}$) exhibited hypersensitivity to FUdR in contrast to other TLS factors, which showed lower FUdR sensitivity than that of the $REV1^{-/-}$ or $REV3^{-/-}$ cells, we hypothesized that Polζ plays other role(s) to prevent cell death upon FUdR in addition to its role in TLS. To test this hypothesis, we monitored the acute

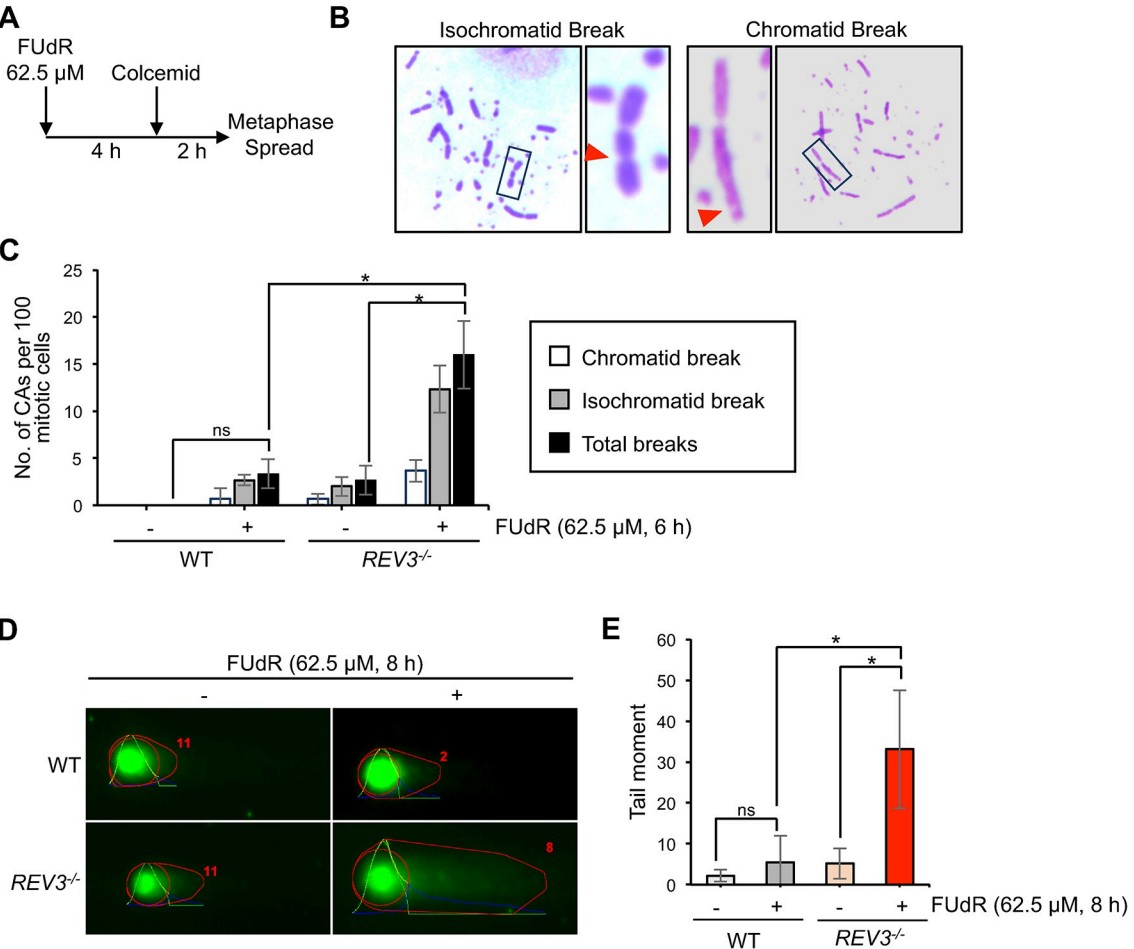

**Fig 3. Measurement of DNA damage induced during FUdR-mediated misincorporation in nascent DNA.** (A) Schematic representation of the experimental protocol to measure chromosomal aberrations. (B) Representative images of chromatid breaks and isochromatid breaks at 6 h after FUdR treatment. Red arrowheads indicate breaks in chromosomes. (C) Number of chromosomal aberrations (CAs)/100 mitotic nuclei in the indicated genotypes. Error bars represent standard deviation. At least 100 mitotic cells were counted for each cell line. *p*-values were calculated by student t-test. ns, not significant; *$p < 0.05$. (D) Representative comet images (upper panel) of REV3 proficient and -deficient cells following treatment with 62.5 uM FUdR for 8 h. (E) The bar graph (upper panel) represents the median and SD of comet tails. n ≥ 100 comets were scored for each data set. *$p < 0.05$ (Student's *t*-test).

effects of FUdR on chromosomes. Analyzing the number of chromosomal aberrations (CAs) in mitotic chromosome spreads 6 h after FUdR exposure enabled the evaluation of the effects of 5-FdUMP and dUMP incorporation into the nascent DNA (Fig 3A). We measured the number of chromosomal aberrations and classified them into isochromatid breaks (breakage of two sister chromatids the same location) and chromatid breaks (breakage of one of the two sister chromatids) (Fig 3B). We found that *REV3⁻/⁻* cells exhibited more CAs than that in the wild-type cells (Fig 3C). To further support this observation, we measured the level of FUdR treatment-induced DNA damage using a comet assay under alkaline conditions to evaluate the extent of DNA lesions, including DSBs and single-strand breaks. We found that *REV3⁻/⁻* cells showed augmented tail moment compared with that of the wild-type cells, indicating increased damage accumulation in REV3-deficient cells (Fig 3D and 3E). Taken together, these results demonstrate that REV3 averts FUdR-induced DNA damage, which suppresses cell death.

## REV3 is required for intra-S checkpoint activation in response to FUdR treatment

Having established the role of REV3 in suppressing DNA damage, we investigated the effect of the loss of REV3 on gross DNA replication by monitoring cell cycle distribution following FUdR treatment. We found that at 3–6 h after FUdR treatment, wild-type cells accumulated cells in the G1 or early S phases, whereas $REV3^{-/-}$ cells did not show such changes in cell cycle distribution (Fig 4A). Moreover, $REV3^{-/-}$ cells exhibited an increased proportion of the sub-G1 fraction (dead cell fraction) 12–24 h after FUdR treatment, which was considerably higher than that observed in wild-type cells (Fig 4A). To distinguish cells in the G1-phase from those in the early S phase, we pulse-labeled the cells with BrdU and found that FUdR treatment caused early S phase arrest in wild-type cells but not in $REV3^{-/-}$ cells (Fig 4B and 4C). The same trend was observed in $REV1^{-/-}$ cells, which proves that the defect of early-S phase arrest is not caused specifically by the loss of REV3 function but is likely to be a functionality of Polζ [15] (S2 Fig). These defects in cell cycle arrest at the early S phase observed in $REV3^{-/-}$ cells were also observed in human TK6 cells (Fig 4D and 4E), suggesting that the role of REV3 in inducing cell cycle arrest at the early S phase is conserved in humans. Collectively, these results show that Polζ plays an evolutionarily conserved role in intra-S checkpoint activation.

## REV3 promotes phosphorylation of Chk1 upon FUdR treatment

As REV3 is critical for intra-S checkpoint activation, we next sought to elucidate the underlying mechanism. Chk1 is a pivotal factor in the intra-S checkpoint, and its activity is modulated by the phosphorylation of its S345 residue [55]. We investigated the phosphorylation status of Chk1 after FUdR exposure using antibodies specific to Chk1-pS345 and found that Chk1 phosphorylation at residue S345 was induced in wild-type DT40 cells after FUdR treatment, indicating that FUdR-mediated DNA damage triggers Chk1-mediated DNA damage checkpoint (Fig 5A and 5B). Notably, Chk1 phosphorylation at S345 substantially reduced in the absence of REV3 in DT40 cells, indicating a critical role for REV3 in the activation of the Chk1-mediated DNA damage checkpoint (Fig 5A and 5B). Similarly, we observed the reduced Chk1 phosphorylation upon FUdR treatment in human TK6 cells (S3 Fig). In contrast, we found that REV3 is not required for Chk1 phosphorylation during replication arrest induced by hydroxyurea (HU) (S3 Fig). Taken together, these data indicate that REV3 promotes Chk1 phosphorylation upon FUdR exposure.

Polζ makes a quaternary complex with REV1 and Polκ [56,57], and former studies revealed that ATR-dependent checkpoint in response to HU, aphidicolin, and UV irradiation is compromised in the absence of either REV1 or Polκ [58,59]. These facts led us to analyze the involvement of Polκ in stimulating intra-S checkpoint upon FUdR treatment (S3 Fig). We found that both $POLH^{-/-}$ and $POLK^{-/-}$ cells are proficient in intra-S checkpoint activation. These results are consistent with the stronger FUdR sensitivity of $REV1^{-/-}$ and $REV3^{-/-}$ cells than that of other TLS mutants including $POLH^{-/-}$ and $POLK^{-/-}$ cells (Fig 1C). These results indicate that Polζ plays a critical role in checkpoint activation upon the replication stress induced by FUdR without requiring Polκ.

## REV3-mediated checkpoint promotes cellular tolerance to FUdR independently of the canonical ATR-Chk1 checkpoint signal pathway

To analyze the functional relationship between the REV3-mediated checkpoint signal and the canonical ATR-Chk1 checkpoint signal pathway, we analyzed the cellular sensitivity of $REV3^{-/-}$ cells to FUdR in the presence of ATR inhibitor (VE821) or Chk1 inhibitor (UCN-01). As

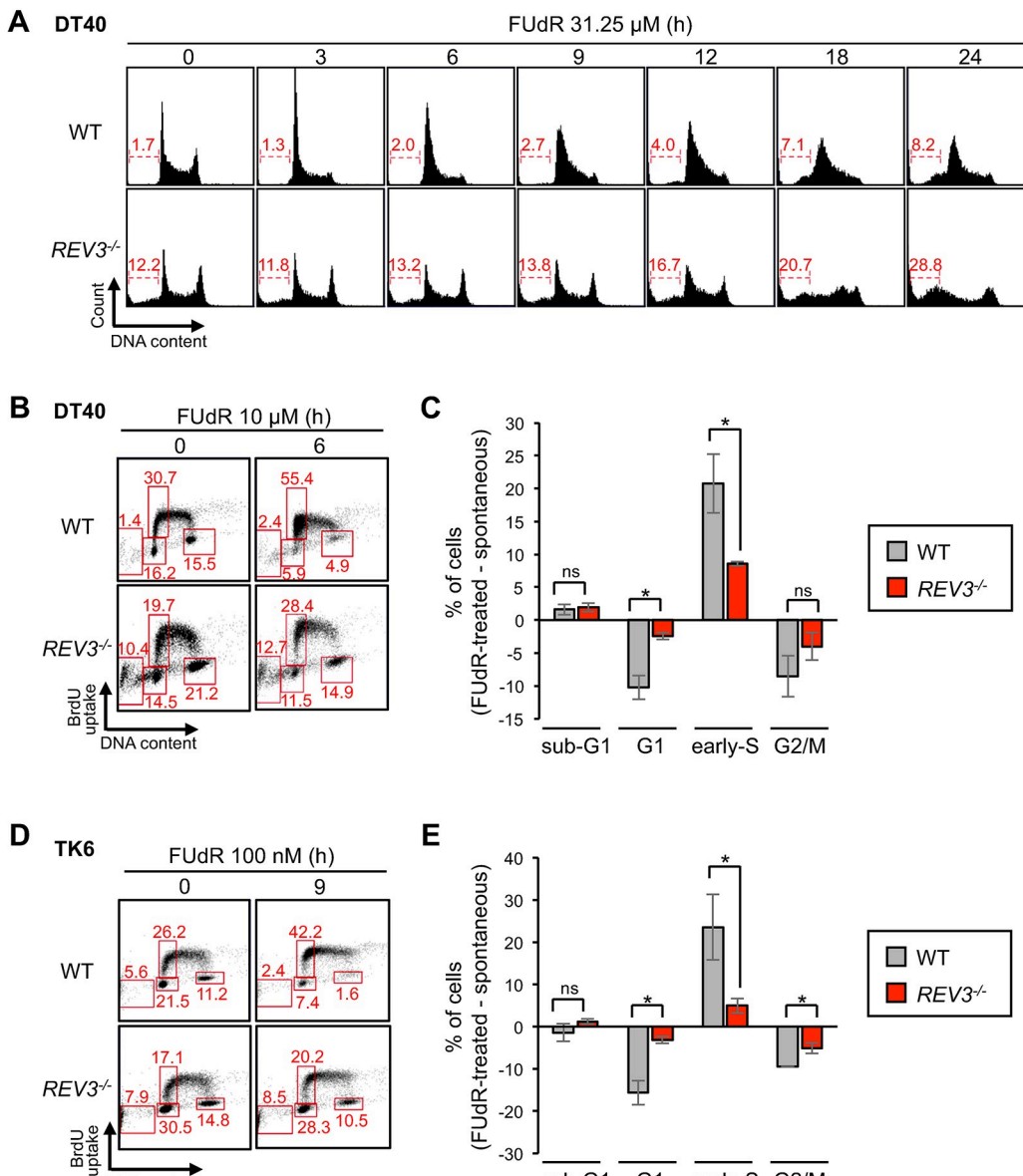

**Fig 4. Involvement of REV3 in intra-S checkpoint activation.** (A) Cells were treated with 31.25 μM FUdR for designated times (0–24 h). The histogram presents the cell cycle distribution. The DNA content (stained using propidium iodide) is displayed on the x-axis on a linear scale and the number of cells counted is presented on the y-axis. Two peaks at 0 h correspond to cells in the G1 (left) and G2 (right) phases. The numbers above brackets represent the percentage of cells in sub-G1 (dead cell fraction). (B) DT40 cells were treated with 10 μM of FUdR for 6 h and pulse-labeled with BrdU for 15 min. Representative cell-cycle distribution for the indicated genotypes. The DNA content (stained using propidium iodide) is displayed on the x-axis on a linear scale, and the BrdU uptake (stained using anti-BrdU antibody) is displayed on the y-axis on a logarithmic scale. The top, lower left, and lower right gates correspond to cells in the early-S, G1, and G2/M phases, respectively. The numbers show the percentage of cells that fall within each gate. (C) The fraction of cells falling in the gate corresponding to the early-S phase and sub-G1 was quantified. Differences in the percentage of cells after FUdR-treatment are shown. Average and SD from three independent experiments are shown. $^*p < 0.05$ (Student's $t$-test). (D) TK6 cells were treated with 100 nM FUdR for 9 h and pulse-labeled with BrdU for 15 mins. Representative cell-cycle distribution for the indicated genotypes (obtained using the same procedure described in (B)). (E) Quantification of the fraction of cells falling in the gate corresponding to early-S phase and sub-G1 is presented (obtained using the same procedure as that described in (C)). Differences in the percentage of cells after FUdR-treatment are shown. Average and SD from three independent experiments are shown. $^*p < 0.05$ (Student's $t$-test).

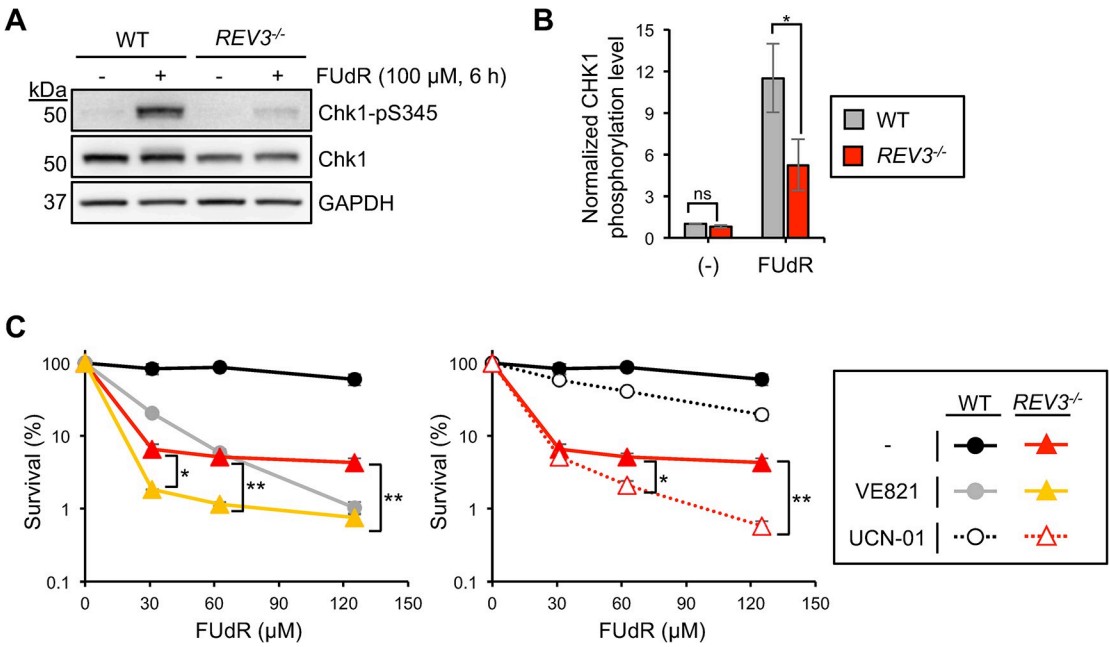

**Fig 5. REV3 promotes Chk1 phosphorylation.** (A) Indicated DT40 cells were treated with 100 µM of FUdR for 6 h. Cell extracts were blotted for Chk1-pS345, Chk1, and GAPDH (loading control). (B) Quantification of Chk1 phosphorylation levels for indicated cells. Chk1-pS345 intensities were quantified and normalized to those of unmodified Chk1. The mean values of three independent experiments and SE were plotted. ns, not significant; $^*p < 0.05$ (Student's $t$-test). (C) DT40 cells with the indicated genotypes were assessed for sensitivity to FUdR in the presence of ATR inhibitor (VE821, 2 µM) or Chk1 inhibitor (UCN-01, 200 nM). Indicated DT40 cells were cultured for 48 h in the presence of an indicated concentration of FUdR. ns, not significant; $^*p < 0.05$; $^{**}p < 0.01$ (Student's $t$-test).

expected, cell viability of wild-type upon FUdR was reduced by the VE821 or UCN-01 (Fig 5C). More importantly, both VE821 and UCN-01 also reduced cellular viability of *REV3*⁻/⁻ cells (Fig 5C). These data suggest that REV3 mediates cellular tolerance to FUdR at least partly independently of the canonical ATR-Chk1 checkpoint signal. As residual Chk1 phosphorylation was observed in *REV3*⁻/⁻ cells (Fig 5A and 5B), Chk1 activation through the ATR axis might promote cellular tolerance to FUdR in parallel to REV3 mediated checkpoint.

### REV3-dependent intra-S checkpoint prevents uracil incorporation

Based on the role of REV3 in intra-S checkpoint activation following FUdR treatment, we hypothesized that intra-S checkpoint-deficient *REV3*⁻/⁻ cells may show increased dUMP- and 5-FdUMP-incorporation. We tested this hypothesis using a modified alkaline comet assay protocol [60], in which cells were treated with the UNG enzyme (UNG-comet) to convert U and FU to abasic (AP) sites that are prone to cleavage by alkaline treatment (Fig 6A). Then, the cells were treated with much lower concentrations of FUdR than those used in the normal alkaline comet assay to minimize the induction of comet formation independent of UNG treatment. In conformance with our assumption, *REV3*⁻/⁻ cells exhibited increased UNG-dependent comet (Fig 6B). Thus, we concluded that REV3 functions in intra-S checkpoint activation to prevent dUMP and 5-FdUMP incorporation upon FUdR treatment.

### Discussion

In this study, the catalytic subunit of Polζ, namely, REV3, was identified as a crucial factor for cellular tolerance to FUdR (Fig 1). Other TLS-deficient mutants also showed increased

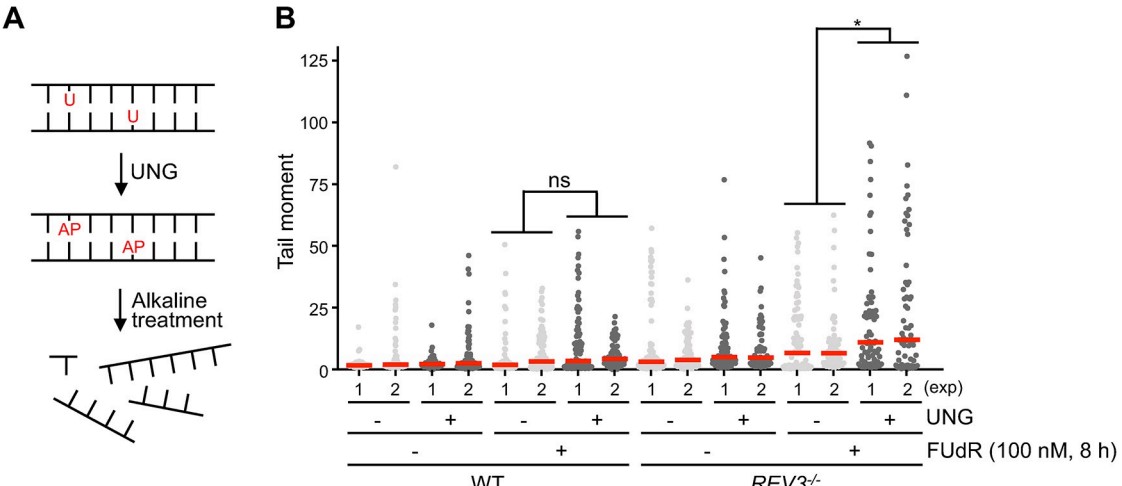

**Fig 6. REV3 prevents incorporation of dUMP and 5-FdUMP in nascent DNA.** (A) Model for the UNG-comet protocol. UNG removes dUMP and 5-FdUMP incorporated in the genome, resulting in the formation of abasic (AP) site. Alkaline treatment induces strand dissociation and breaks at the AP site. (B) Dot plots represent comet tail moment/cell and the red bars display the median (n $\geq$ 100 cells per condition). The experiment was repeated twice (exp1 and exp2). $p$-values were calculated by Mann–Whitney–Wilcoxon test. ns, not significant; $^*p < 0.05$.

sensitivity to FUdR compared with that of wild-type cells (Fig 1). Moreover, $REV3^{-/-}$ cells exhibited slowed replication fork progression on the FUdR-damaged template strand (Fig 2). These data indicate a role for the TLS pathway in cellular tolerance to FUdR. $REV3^{-/-}$ cells exhibited an increased number of CAs and augmented tail moments during dUMP and 5-FdUMP incorporation following FUdR treatment (Fig 3). These data demonstrate the role of REV3 in suppressing FUdR-induced DNA damage. Moreover, we found that REV3 activates the intra-S phase checkpoint, thereby restricting the incorporation of dUMP and 5-FdUMP into the genome (Figs 4–6).

The phenotypes of the human TK6 and chicken DT40 $REV3^{-/-}$ mutants were similar following FUdR treatment (Fig 1), indicating that the roles of REV3 in preventing cell death upon FUdR have been evolutionarily conserved. DT40 cells showed a more prominent phenotype than TK6 cells (Fig 1), potentially due to their lack of functional p53 [61], as well as increased replication stress caused by the overexpression of c-Myc [62].

We showed that the TLS function of REV3 is responsible for cellular tolerance to FUdR (Figs 1–2). This observation is consistent with the previous report showing the involvement of REV7, the non-catalytic subunit of Polζ, in the cellular tolerance to 5-FU [63]. Moreover, we previously showed that Polζ plays a critical role in the tolerance to the overexpression of activated induced deaminase (AID), which converts cytosine into uracil [14]. Mechanistically, formed uracil is further processed by glycosylases creating abasic sites on which Polζ plays a critical role. These observations are consistent with our current data that the TLS function of REV3 maintains replication fork progression on dUMP- and 5-FdUMP-incorporated damaged templates (Fig 2).

$REV3^{-/-}$ cells exhibited an increased number of CAs (Fig 3B). We previously reported that interference of DNA replication can induce CAs unassociated with DSBs but associated with single strand gaps [40], and it is possible that FUdR-induced CAs in $REV3^{-/-}$ cells are unassociated with DSBs. Should this be the case, such CAs might be caused by unreplicated single strand gaps. This might be indeed the case, since we showed that an augmented level of tail moments in $REV3^{-/-}$ cells in alkaline conditions compared with that in wild-type cells (Fig 3C).

We demonstrated that REV3 plays a role in Chk1 phosphorylation (S345)-mediated intra-S-phase checkpoint activation upon FUdR incorporation-mediated replication arrest (Figs 4–5). Since we observed a slight decrease in Chk1 protein levels in DT40 *REV3*$^{-/-}$ cells (Fig 5A) and TK6 *REV3*$^{-/-}$ cells (S3 Fig), it is possible that REV3 contributes to the faithful checkpoint activation through maintaining Chk1 protein stability at least partly. Moreover, our data suggest that this REV3-mediated mechanism was not involved in HU-mediated checkpoint activation (S3 Fig). As most previous studies have investigated DNA replication stress signaling using HU-mediated replication arrest [64,65], REV3-mediated Chk1 activation by FUdR may have been overlooked. Since HU interferes with the synthesis of all nucleotides leading to the arrest of the DNA synthesis process and generation of long stretches of ssDNA [22], such HU-mediated long stretches of ssDNA might efficiently activate canonical ATR-Chk1 pathway. In contrast, FUdR interferes with the synthesis of the dTTP nucleotide and also promotes dUTP accumulation [28–33], and then resulting in the discontinuous DNA synthesis at all replisomes generating single stranded (ss)DNA gaps. Since the 9-1-1 alternative checkpoint clamp comprising Hus1, Rad1, and Rad9 physically interacts with REV7 [66], a possible rationale is that Polζ recruits 9-1-1 complex to such ssDNA gaps to transmit DNA damage signal to Chk1. This view is consistent with our observations that REV3-mediated DNA damage checkpoint activation acts in parallel with the canonical ATR-Chk1 pathway (Fig 5C). Further study is required to understand the pivotal relationship between Polζ (REV3, REV7), 9-1-1, and ATR in the FUdR-activated intra-S-phase checkpoint.

We demonstrated that *RAD18*$^{-/-}$ cells exhibited significantly higher sensitivity to FUdR than *PCNA*$^{-/K164R}$ cells (Fig 1C). Since Rad18 induces PCNA ubiquitination at K164 and activates TLS [53,67], these different FUdR-sensitivities between *RAD18*$^{-/-}$ and *PCNA*$^{-/K164R}$ cells were unexpected and suggested possible roles of Rad18 besides its role to activate TLS. A possible rationale is that Rad18 might avoid the toxic NHEJ leading to chromosome fusion. As FUdR affects DNA replication (Fig 2), DSBs might be induced due to fork collapse and such DSBs can be repaired by either HR or NHEJ. But one-ended DSB caused during DNA synthesis should be exclusively repaired by HR and NHEJ-mediated repair leading to detrimental chromosome fusion is strictly suppressed by Rad18 [35]. Another possible hypothesis is that Rad18 contributes to cellular tolerance to FUdR by inducing break-induced replication (BIR)-like DNA synthesis at the stalled replication fork [68]. Rad18 is recruited to ubiquitinated histone H2AX upon HU and induces DNA synthesis in a manner that requires BIR factors including PolD3 [68]. This view was supported by the observation that *POLD3*$^{-/-}$ cells exhibited higher FUdR-sensitivity than *PCNA*$^{-/K164R}$ cells like *RAD18*$^{-/-}$ cells (Fig 1C).

Suppression of REV3 expression is observed in the early stages of tumorigenesis in several cancers, including colon and prostate cancers [66,69]. The state of the *REV3* gene, including its expression state and mutations, may be a helpful biomarker for predicting the efficacy of FUdR-based cancer chemotherapy. In conclusion, this study presents a previously unknown role of REV3 in FUdR tolerance, wherein REV3 promotes cellular tolerance to FUdR through TLS and intra-S checkpoint activation. We believe that these findings can expedite the development of cancer chemotherapy based on data obtained from the transcriptome and genomic mutation of cancer cells.

## Methods and materials

### DT40 cell culture and cellular-sensitivity analysis

The DT40 and TK6 cell lines used in this study are listed in Table 1. Culture conditions and cellular sensitivity analyses (ATP assay) were performed according to previously described methods [9,70,71]. The DT40 cells were maintained at 39.5˚C in a humidified atmosphere

with 5% $CO_2$. They were cultured in DMEM/F-12 medium (GIBCO-BRL, Grand Island, NY, USA) supplemented with 10% heat-inactivated fetal bovine serum (FBS) (AusgeneX, lot No. QLD 4210), 1% chicken serum (GIBCO-BRL, Grand Island, NY, USA), 50 μM mercaptoethanol (Invitrogen, MA, USA), L-glutamine (Nacalai Tesque, Kyoto, Japan), 50 U/mL penicillin, and 50 μg/mL streptomycin (Nacalai Tesque). TK6 cells were maintained at 37°C in a humidified atmosphere with 5% $CO_2$, and cultured in Roswell Park Memorial Institute 1640 medium (Nacalai Tesque) supplemented with sodium pyruvate (1.8 mM), L-glutamine (2 mM), penicillin (50 U/mL), streptomycin (50 μg/mL; Nacalai Tesque), and 5% heat-inactivated horse serum (Gibco). For the ATP assay, actively growing cells were treated with FUdR. In this assay, 96-well plates were used to treat 800 cells with the indicated concentration of FUdR in 80 μL of medium, followed by incubation at 39.5°C for 48 h. According to the manufacturer's instructions, 100 μL of the cell-containing medium was added to 96-well plates, to which Cell-Titer-Glo (Promega, WI, USA) was added to quantify the ATP content. Fluoroskan Ascent FL (Thermo Fisher Scientific Inc., Waltham, MA, USA) was used to measure the luminosity. $LD_{90}$ values indicating the drug concentration that reduces cellular survival by 10% are calculated from the survival curves shown in S1 Fig by GraphPad Prism 9 software which can estimate $LD_{90}$ by extrapolating survival curves if the data do not include 10% survival data.

## Genotoxic drugs and inhibitors used in this study

Floxuridine (FUdR; TCI, Tokyo, Japan), Hydroxyurea (HU; Nacalai Tesque), the ATR inhibitor, VE821 (Funakoshi, Tokyo Japan), and the Chk1 inhibitor, UCN-01 (Funakoshi, Tokyo Japan) were used.

## Colony survival assay

A cell survival assay using TK6 cells was performed as a fraction of surviving colonies. A colony-formation assay was performed as described previously [26]. FUdR sensitivity analysis was performed by plating cells into DMEM-ham's F12 (GE, MA, USA) medium containing 1.5% of methylcellulose, 10% heat-inactivated horse serum (HS), sodium pyruvate (1.8 mM), L-glutamine (2 mM), penicillin (50 U/mL), and streptomycin (50 μg/mL along with the indicated concentration of FUdR) and counting the number of surviving colonies after incubation for 14 days at 37°C in a humidified atmosphere with 5% $CO_2$.

## Chromosome aberration analysis

Chromosomal aberrations (CAs) were analyzed as previously described [9,70,71]. DT40 cells were treated with 62.5 μM of FUdR for 6 h and with 0.1 μg/mL colcemid (GIBCO) for the last 2 h to arrest the cells in the M phase. The cells were pelleted through centrifugation (1200 rpm for 5 min), resuspended in 75 mM KCl (10 mL) for 13 min at 23°C, and fixed in a freshly prepared 3:1 mixture (2 mL) of methanol and acetic acid (Carnoy's solution). The pelleted cells were resuspended in Carnoy's solution (7 mL), dropped onto cold glass slides, and air-dried. The slides were stained with 5% HARLECO Giemsa Stain solution (Nacalai Tesque) for 10 min, rinsed with water and acetone, and dried at 20°C. The slides were examined under an ECLIPSE-Ni microscope (NIKON, Tokyo, Japan) at 1000 × magnification. The chromosomes within each mitotic cell were scored.

## Single-cell gel electrophoresis (alkaline comet assay, UNG comet assay)

Chicken DT40 cells were treated with 62.5 μM of FUdR for 8 h at 39.5°C, and the tail DNA moments [72] that reflect the induced damages were measured. Alkaline comet and single-cell

gel electrophoresis assays were performed as described previously [26]. For the UNG comet assay, the cells ($3 \times 10^4$ cells/slide) were resuspended in 100 µl of 1% low-melting agarose (Sigma-Aldrich, WI, USA) in PBS and spread onto microscopy slides that were previously coated with 1% agarose (Bio-Rad, CA, USA). The cells were lysed in the lysis solution (2.5 M NaCl, 100 mM $Na_2$EDTA, 10 mM Tris-base, 8 g/L NaOH to maintain pH at 10; 1% Triton X-100 and 0.5% N-lauroylsarcosine sodium salt were added prior) at 4˚C for 2 h. Each slide was washed thrice for 5 min in uracil DNA glycosylase (UNG) buffer (40 mM HEPES, 100 mM KCl, 0.5 mM EDTA, and 0.1% BSA; pH was adjusted to 8.0 using KOH). Then, 100 µL of either UNG buffer with DNA glycosylase enzyme (2 U/mL, Roche, Basel, Switzerland) or UNG buffer alone were added, and the slides were covered with a glass coverslip. The slides were incubated in a moist atmosphere at 37˚C for 1 h. After enzyme treatment, the slides were placed in a tank with cold running buffer (300 mM NaOH, 1 mM $Na_2$ EDTA, pH was adjusted to 13 using HCl) for 40 min before electrophoresis for 90 min at 25 V. Comet Analysis System OpenComet [73] was used to quantify tail DNA moments in DT40 cells. 150 cells were scored per sample.

## DNA fiber assay

DNA fiber analysis was performed as previously described with minor adjustments [74]. Following a 25 µM chlorodeoxyuridine (CldU; Sigma-Aldrich) pulse, the cells ($5 \times 10^5$ in 1 mL of medium) were pulse-labeled with 250 µM iododeoxyuridine (IdU; Sigma-Aldrich). Glass slides were coated with the reconstituted cells suspended in ice-cold PBS and tilted to spread DNA after the cells were lysed using DNA fiber lysis buffer (0.5% SDS, 200 mM Tris-HCl [pH 7.4], and 50 mM EDTA). The glass slides were fixed through immersion in Carnoy's solution (MeOH:AcOH, 3:1) for 3 min and in 70% EtOH for 1 h, followed by washing with PBS for 3 min and immersion in HCl (2.5 N) for 30 min to denature the DNA molecules, and neutralization with 0.1 M sodium tetraborate for 3 min. After washing with PBS, the slides were incubated with rat anti-BrdU antibodies (1:200; Abcam) and mouse anti-BrdU antibodies (1:50; BD Biosciences, NJ, USA), to detect anti-CldU and anti-IdU reactions, respectively. The secondary antibodies used were Alexa Fluor 488 anti-mouse IgG (1:100; Invitrogen) and Cy3-conjugated anti-rat IgG (1:400; Jackson ImmunoResearch Laboratories, PE, USA). The primary and secondary antibodies were incubated for 1 h each at room temperature, followed by washing with PBS containing 0.05% Tween 20. PermaFluor mounting medium was used to mount coverslips (Lab Vision, CA, USA). ImageJ software was used to measure the fiber lengths.

## Flow cytometric analysis of cell cycle distribution

After fixing the pretreated cells for 30 min at 4˚C in chilled 70% ethanol, they were stained at 37˚C for 30 min with ribonuclease A (50 µg/mL; Sigma-Aldrich) and PI (5 µg/mL; Nacalai Tesque) in the presence of 1% bovine serum albumin (BSA; Nacalai Tesque).

BrdU FACS was performed as previously described [4]. The cells were cultured in media with or without 10 µM of FUdR for 6 h for DT40 cells and 100 nM FUdR for 9 h for TK6 cells, followed by labeling with 20 µM BrdU for the last 15 min. The cells were harvested and fixed overnight in 70% ethanol at 4˚C, followed by successive incubation as follows: (i) in 2N HCl and 0.5% Triton X-100 for 30 min at room temperature; (ii) in mouse anti-BrdU antibody (1:100; BD Biosciences, NJ, USA) for 60 min at room temperature; (iii) in FITC-conjugated anti-mouse antibody (1:50; Southern Biotech, Birmingham, UK) for 30 min at room temperature; (iv) in 5 µg/mL PI in PBS. Flow cytometric analysis was performed using a BD Accuri C6 flow cytometer (BD Biosciences).

## Western blotting

Western blotting was performed to detect phosphorylated Chk1 as previously described [51]. Whole-cell extracts were prepared by lysing cells directly with $1 \times$ Laemmli buffer (50 mM Tris-HCl, 2% SDS, 10% glycerol, 100 mM dithiothreitol, and 2.5 mg/mL bromophenol blue) and boiling for 15 min. The protein samples were separated using electrophoresis on 5–15% gradient precast gel (Fuji film, Tokyo, Japan) with SDS running buffer. Proteins were then transferred onto PVDF membrane (GE, MA, USA) using $1 \times$ transfer buffer (25 mM Tris-HCl, 192 mM glycine, 20% methanol, and 0.01% SDS) at 30 V overnight in a wet transfer tank (Bio-Rad). After transfer, the membranes were saturated for 30 min using PBS-T (0.05% Tween 20) containing 5% skim milk. After washing with PBS-T, the membranes were incubated with primary antibodies for 1 h at room temperature. Then, the membranes were washed with PBS-T and incubated with the secondary antibodies for 40 min at room temperature. After washing with PBS-T, the protein signals were detected using ImmunoStar$^R$ LD (Fuji Film). Primary antibodies; anti-CHK1 (1:500; sc-8408, Santa Cruz Biotechnology, SCBT, CA, USA), anti-CHK1-pS345 (1:1000; #2341, Cell Signaling Technology, CST, MA, USA), anti-GAPDH (1:1000; sc-32233, SCBT).

## Supporting information

**S1 Fig. REV3 plays pivotal roles in cellular tolerance to FUdR.** DT40 cells of the indicated genotypes were assessed for FUdR sensitivity. The cells were cultured for 48 h in the presence of FUdR at the indicated concentrations. The x-axis indicates the dose of FUdR on a linear scale, whereas the y-axis indicates the percentage of cell survival on a logarithmic scale. Error bars represent the standard deviation of three independent experiments. ns, not significant; $^*p < 0.05$; $^{**}p < 0.01$ (Student's $t$-test).
(PDF)

**S2 Fig. REV1 and REV3 are similarly required for FUdR-mediated intra-S phase checkpoint activation.** Indicated DT40 cells were treated with 31.25 μM FUdR for designated times (0–6 h). The histogram presents the cell cycle distribution. The DNA content (stained with propidium iodide) is displayed on the x-axis on a linear scale, and the number of cells counted is presented on the y-axis. Two peaks at 0 h correspond to cells in the G1 (left) and G2 (right) phases. The numbers above brackets represent the percentage of cells in sub-G1 (dead cell fraction).
(PDF)

**S3 Fig. REV3 promotes Chk1 phosphorylation.** (A) Indicated TK6 cells were treated with 250 nM of FUdR or 50 μM of hydroxyurea (HU) for 9 h. Cell extracts were blotted for Chk1-pS345, Chk1, and GAPDH (loading control). (B) Quantification of Chk1 phosphorylation levels for indicated cells. Chk1-pS345 intensities were quantified and normalized to those of unmodified Chk1. The mean values of three independent experiments and SE were plotted. ns, not significant; $^*p < 0.05$; $^{**}p < 0.01$ (Student's $t$-test). (C) Indicated DT40 cells were treated with 31.25 μM FUdR for designated times (0–6 h). The histogram presents the cell cycle distribution. The DNA content (stained with propidium iodide) is displayed on the x-axis on a linear scale, and the number of cells counted is presented on the y-axis. Two peaks at 0 h correspond to cells in the G1 (left) and G2 (right) phases. The numbers above brackets represent the percentage of cells in sub-G1 (dead cell fraction).
(PDF)

**S1 Data. All raw data gained in this study are presented in this supplementary information.**
(XLSX)

## Acknowledgments

The authors would like to thank Dr. Toyofumi Yamaguchi for sharing the reagents, Ms. Minami Fukuchi for her assistance in deriving the preliminary data. We thank all members of Hirota laboratory for their helpful discussion. We are grateful to the Radioisotope Research Center of Tokyo Metropolitan University for their assistance with isotope usage.

## Author Contributions

**Conceptualization:** Kouji Hirota.

**Formal analysis:** Mubasshir Washif, Ryotaro Kawasumi.

**Funding acquisition:** Ryotaro Kawasumi, Kouji Hirota.

**Investigation:** Mubasshir Washif, Ryotaro Kawasumi.

**Project administration:** Kouji Hirota.

**Resources:** Kouji Hirota.

**Supervision:** Kouji Hirota.

**Writing – original draft:** Mubasshir Washif.

**Writing – review & editing:** Ryotaro Kawasumi, Kouji Hirota.

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
