## [Decision Letter · Decision Letter 0]

21 Mar 2024

Dear Dr Hirota,

Thank you very much for submitting your Research Article entitled 'REV3 promotes cellular tolerance to 5-fluorodeoxyuridine by activating translesion DNA synthesis and intra-S checkpoint' to PLOS Genetics.

The manuscript was fully evaluated at the editorial level and by independent peer reviewers. The reviewers appreciated the attention to an important problem, but raised some substantial concerns about the current manuscript. Based on the reviews, we will not be able to accept this version of the manuscript, but we would be willing to review a much-revised version. We cannot, of course, promise publication at that time.

Should you decide to revise the manuscript for further consideration here, your revisions should address the specific points made by each reviewer. We will also require a detailed list of your responses to the review comments and a description of the changes you have made in the manuscript. It would be also helpful for consideration of revised version to have a copy of revised manuscript with tracked or marked up changes

If you decide to revise the manuscript for further consideration at PLOS Genetics, please aim to resubmit within the next 60 days, unless it will take extra time to address the concerns of the reviewers, in which case we would appreciate an expected resubmission date by email to plosgenetics@plos.org.

We are sorry that we cannot be more positive about your manuscript at this stage. Please do not hesitate to contact us if you have any concerns or questions.

Yours sincerely,

Dmitry A. Gordenin, Ph.D.

Academic Editor

PLOS Genetics

Gregory Barsh

Editor-in-Chief

PLOS Genetics

Reviewer's Responses to Questions

**Comments to the Authors:**

Reviewer #1: Review of: “REV3 promotes cellular tolerance to 5-fluorodeoxyuridine by activating translesion DNA synthesis and intra-S checkpoint”, by M. Wahif et al., submitted to PLOS Genetics.

The authors administer 5-fluorodeoxyuridine (5-dFUdR), a well-known anticancer an antiviral agent, to chicken or human cells and observe that cell variants lacking a functional REV3 enzyme survive more poorly and display several other defects relating to DNA replication, such as fork movement slowdown, chromosomal breaks and other DNA aberrations, altered cell-cycle progression, and loss of S1 checkpoint control. Overall, the authors conclude that REV3, encoding the catalytic polymerase unit of the established Translesion DNA-Synthesis polymerase Pol Zeta, plays a critical role not only in translesion synthesis but also in cell-cycle control and checkpoint activation after induction of DNA damage.

Overall, this work appears well performed and meaningfully interpreted. I have some minor concerns primarily on issues of presentation and explanation, as outlined below:

1. The section on cell-cycle progression (pages 8-9 and Fig. 4) is hard to follow, especially for readers not directly familiar with cell sorting, etc. In Fig. 4A, what are the peaks, what are the read numbers above the brackets? (same for Fig. S2) Simply referring to ref. 4 for BrdU cell sorting is not sufficient. Needs better description.

2. When you start exposing cells to FUdR what is the state of the cells? Are they actively growing? Please how cells were prepared for treatment.

3. Are the results shown in Fig. 1 from plating experiments for surviving colonies? What did you use the ATP assay for? Please explain.

4. The FUdR sensitivity in Fig. 1A is based on a comparison of 90% survival data (Results first paragraph). However, looking at the actual survival data there are very few data point for this level of survival, at least for the most sensitive cells. Please explain how you did this.

5. In Fig. 3B, explain the purpose of the red pointing triangles.

6. In fig. 5C, VE821 should be BE821 (or reverse)?

7. In Fig. 6, what are (exp) 1, 2 (are they duplicates? Please explain)

8. Overall, the authors make strong statements on the role of REV3 in checkpoint activation for the case of FUdR. In view of certain effects that REV3 deletion seems to have on cell cycle events even in unexposed cells (various panels of Fig. 4), maybe some moderation may be appropriate. Also, one might comment on these effects as well. Might be relevant, as some of the effects in the deletion mutants could be indirect.

9. In at least one case (page 14) I saw mMolar instead of microMolar. Please check carefully.

Reviewer #2: The authors analyzed the mechanism underlying cellular resistance to FUdR. They propose that REV3 promotes cellular tolerance to FUdR by activating TLS and the intra-S checkpoint. In particular, the conclusion regarding the intra-S checkpoint is unexpected. These data, revealing the mechanism of such an important biological reaction, should significantly contribute to this field. Furthermore, researchers not only in DNA repair (in Genetics) but also in the field of cancer research might be interested in this manuscript. Below are my suggestions for further strengthening the manuscript.

Major:

1) REV3-/- DT40 cells showed considerably high sensitivity to FUdR (Figures 1A and C). However, REV3-/- TK6 cells exhibited only mild sensitivity to FuDR (Figure 1B). Thus, the effect of depleting Rev3 on FUdR sensitivity is different in DT40 cells compared to TK6 cells. The authors should discuss the reasons for this difference.

2) In Figure 2, the rate of DNA synthesis in REV3-/- cells was significantly slower compared to that of wild-type cells after incubation with FUdR for 12 hours. Consequently, the authors concluded that REV3-mediated TLS maintains replication fork progression in the presence of 5-FdUMP and dUMP on template strands. However, the authors should exercise caution in interpreting these data. Since REV3-/- cells exhibit increased incorporation of dUMP and 5-FdUMP (Figure 6), it is challenging to attribute replication fork progression solely to the effect of TLS.

3) In Figure 3, REV3-/- cells exhibited more chromosome aberrations than the wild-type cells after treatment with FuDR. Additionally, Dr. Hirota reported that replication-blocking agents can cause chromosomal breaks unassociated with DSBs, specifically single-strand gaps left unreplicated (Ref40). Consequently, the authors need to discuss whether the chromosome aberrations induced by FuDR represent DSBs or unreplicated single-strand gaps.

4) In Figures 4 and 5, it is noteworthy that Rev3 contributes to the activation of the intra-S checkpoint induced by FuDR. Conversely, Rev3 is not implicated in the activation of the intra-S checkpoint induced by HU in TK6 cells. The authors should verify the reproducibility of these findings using DT40 cells. Additionally, please discuss the different mechanisms of intra-S checkpoints induced by HU and FuDR, based on the results obtained by the authors.

Minor:

1) In Figure 3B, the image does not distinguish between isochromatid breaks and chromatid breaks. Therefore, it would be preferable to provide a clearer image. If presenting it clearly is challenging, it might be beneficial to include an illustration as well (e.g., Figure 1D and 4B in PMID: 21464321 [Yamamoto et al., PNAS 2010]).

2) On page 10, line 3: "FudR" is incorrect.

3) In Figures 4C and E, please conduct statistical analysis.

4) Is the UNG-comet method novel? If this method has already been developed, please provide a citation.

Reviewer #3: Washif et al., provide evidence for a role of the translesion DNA synthesis polymerase REV3, a subunit of the TLS pol zeta heterodimer, in tolerance to 5-fluorodeoxyuridine(5FdU) in chicken DT40 cells. They identified Rev3 as important to tolerate 5FdU by analyzing a series of cell lines knock out for several genes involved in several DNA repair and DNA damage tolerance pathways. Rev3-/- cells stood out as the most sensitive cells to 5FdU. Authors showed that Rev3-/- cells reduce replication fork speed compared to wild-type cells upon 5FdU treatment and that these cells also were deficient in activating the ATR-dependent checkpoint thus entering mitosis with stretches of unreplicated DNA, resulting in chromosome breakage. These studies are consistent with the role of TLS in tolerating the insertion of 5FdU as previously shown by the authors for the second Pol zeta subuinit REV7, however a role for REV3 in activating the ATR-dependent checkpoint was not previously reported and constitutes the most original part of this work which is of general interest. However, this part is not much developed, and to my opinion, it warrants further investigation in clarifying the role of REV3 in ATR-dependent checkpoint activation, and obtaining some evidence that a similar process may also occur in a mammalian cell line. Below are the main key points that if addressed in my opinion would make this work a strong candidate for publication.

1. While DT40 chicken cells are an interesting and pertinent genetic model, for this work to be of general interest, it would be important to determine whether also mammalian cells knock out for REV3 show a similar sensitivity to 5FdU, compared to knock out of at least another TLS pol.

2. It is known that Pol zeta makes a quaternary complex with Rev1 and Polk (Wojtaszek J et al., JBC 2012; Xie et al., Protein Cell 2012). Interestingly, two previous reports have shown that activation of the ATR-dependent checkpoint is compromised in the absence of either Polk or Rev1 in both Xenopus egg extracts and mammalian cells (Bétous et al., EMBO J 2013; DeStephanis et al., BBRC 2015). Hence, one may wonder whether the inability to phosphorylate Chk1 in REV3-/- cells could be due to destabilization of this complex and/or reduced gene expression of ReV1 or Polk. Also, Fig. 5A shows that the total level of Chk1 is also reduced upon REV3 knock out. A more recent report has provided evidence for Chk1 stabilization by Polk (Dall’Osto et al., MCB 2021). Hence, I believe that this point required further investigation in order to clarify whether the checkpoint activation function of Rev3 may be in concert with that of Rev1 and/or Polk.

3. Another important point is to speculate how 5FdU is dealt with in DT40 cells. Authors show that Rev3-/- cells are much more sensitive to 5FdU than cells bearing a point mutation in the PCNA lysine residue (K164R mutation) ubiquitinated by Rad18, that allows Y-family TLS pols recruitment. Now, neither Polzeta nor Rev1 require PCNAmUb to be recruited. However, Rad18-/- cells are much more sensitive to 5FdU than PCNAk164R. How the authors explain this difference? Then, in terms of TLS function, what can be draw from these results? Is 5FdU TLS on the fly in DT40 cells? This could explain results shown in Fig. 2. Fiber spreading assay in mammalian cells Rev3-/- could be interesting to determine whether what observed by the authors is specific to DT40 cells. In fact, “on the fly TLS” by Rev1 has only been observed in DT40 cells (Edmunds et al., Mol Cell 2008).

Other points

Figure 1, 4B-E, 5C and S1 miss significance tests. Fig. S1 is very crowed, maybe it can be split in several panels so to better read and ascertain the sigbnificance of the differences.

**Have all data underlying the figures and results presented in the manuscript been provided?**

Reviewer #1: Yes

Reviewer #2: Yes

Reviewer #3: Yes

PLOS authors have the option to publish the peer review history of their article (what does this mean?). If published, this will include your full peer review and any attached files.

Reviewer #1: No

Reviewer #2: No

Reviewer #3: **Yes: **Domenico Maiorano

---

## [Decision Letter · Decision Letter 1]

3 Jun 2024

Dear Dr Hirota,

Thank you very much for submitting your Research Article entitled 'REV3 promotes cellular tolerance to 5-fluorodeoxyuridine by activating translesion DNA synthesis and intra-S checkpoint' to PLOS Genetics.

The manuscript was fully evaluated at the editorial level and by independent peer reviewers. Two reviewers do not have additional requests to your submission.

One reviewer still identified some concerns about presentation, interpretations and discussion that we ask you address in a revised manuscript and in your point-by-point response. Your revisions and response should address the specific points made by the reviewer describing modifications and the underlying rationale.

Yours sincerely,

Dmitry A. Gordenin, Ph.D.

Academic Editor

PLOS Genetics

Gregory Barsh

Section Editor

PLOS Genetics

Reviewer's Responses to Questions

**Comments to the Authors:**

Reviewer #1: Comments were addressed appropriately.

Reviewer #2: The authors thoroughly addressed all concerns. I recommend to publish this paper in PLOS Genetics.

Reviewer #3: While the authors have now improved the manuscript, there are still few issues that need to be resolved in order to strengthen the conclusions, listed here below.

1. It is clear from figure 5 and Figure S3 that REV3 deletion affects the stability of Chk1. This observation has to be taken into account in the interpretation of the results.

2. In response to the different sensitivity of PCNAk164R mutant cells compared to Rad18-deleted cells, although its is known that Rad18 is also involved in DNA double strand break repair, I do not see how this function fits with the sensitivity to 5FdU. Maybe the authors think to a possible role of Rad18 in repairing gaps generated by 5FdU in REV3-deleted cells by promoting homologous recombination? (See also comments below). Further, authors have not provided explanation nor additional data about “on the fly” TLS in the presence of 5FdU.

3. Figure S1 still misses statistical significance tests.

4. Discussion, authors provide an explanation for the different effect of 5FdU compared to hydroxyurea (HU) treatment which is unclear to me. In particular I do not see what is the difference between “the entire replisome” and “individual replisomes” as mentioned by the authors. Maybe the authors intend that 5FdU does not arrests all replication forks? If this is the case, I do not see why. As far as I am aware, 5FdU as a milder effect than HU on nucleotide synthesis, since it interferes with the synthesis of the dTTP nucleotide and also promotes dUTP accumulation. This probably generates discontinuous DNA synthesis at all replisomes generating single stranded (ss)DNA gaps. In contrast, HU interferes with the synthesis of all nucleotides leading to arrest of the DNA synthesis process and generation of long stretches of ssDNA since the DNA helicase component of the replisome is not affected. The canonical ATR-Chk1 pathway does depends upon the 9-1-1 complex for activation. Probably the more likely explanation is that Chk1 activation in REV3-deleted cells occurs at ssDNA gaps that are left behind the replication fork? Here the authors discover that REV3 is important to activate the ATR-dependent checkpoint, probably by recruiting the 9-1-1 complex at ssDNA post-replicative DNA gaps.

**Have all data underlying the figures and results presented in the manuscript been provided?**

Reviewer #1: Yes

Reviewer #2: Yes

Reviewer #3: Yes

PLOS authors have the option to publish the peer review history of their article (what does this mean?). If published, this will include your full peer review and any attached files.

Reviewer #1: No

Reviewer #2: No

Reviewer #3: No

---

## [Editor Report · Decision Letter 2]

13 Jun 2024

Dear Dr Hirota,

We are pleased to inform you that your manuscript entitled "REV3 promotes cellular tolerance to 5-fluorodeoxyuridine by activating translesion DNA synthesis and intra-S checkpoint" has been editorially accepted for publication in PLOS Genetics. Congratulations!

Yours sincerely,

Dmitry A. Gordenin, Ph.D.

Academic Editor

PLOS Genetics

Gregory Barsh

Section Editor

PLOS Genetics

Comments from the reviewers (if applicable):

**Data Deposition**

http://datadryad.org/submit?journalID=pgenetics&manu=PGENETICS-D-24-00189R2

**Press Queries**

---

## [Editor Report · Acceptance letter]

27 Jun 2024

PGENETICS-D-24-00189R2 

REV3 promotes cellular tolerance to 5-fluorodeoxyuridine by activating translesion DNA synthesis and intra-S checkpoint 

Dear Dr Hirota, 

We are pleased to inform you that your manuscript entitled "REV3 promotes cellular tolerance to 5-fluorodeoxyuridine by activating translesion DNA synthesis and intra-S checkpoint" has been formally accepted for publication in PLOS Genetics! Your manuscript is now with our production department and you will be notified of the publication date in due course.

With kind regards,

Jazmin Toth

PLOS Genetics

On behalf of:
